# Effects on Muscular Activity after Surgically Assisted Rapid Palatal Expansion: A Prospective Observational Study

**DOI:** 10.3390/bioengineering9080361

**Published:** 2022-08-03

**Authors:** Marco Farronato, Davide Farronato, Aldo Bruno Giannì, Francesco Inchingolo, Ludovica Nucci, Gianluca Martino Tartaglia, Cinzia Maspero

**Affiliations:** 1Department of Orthodontics, Faculty of Medicine, University of Milan, 20142 Milano, Italy; gianluca.tartaglia@unimi.it (G.M.T.); cinzia.maspero@unimi.it (C.M.); 2School of Medicine and Surgery, University of Insubria, Via G. Piatti 10, 21100 Varese, Italy; davide@farronato.it; 3Facial Surgery and Dentistry Fondazione IRCCS Cà Granda, UOC Maxillo, Ospedale Maggiore Policlinico, 20142 Milan, Italy; aldo.gianni@unimi.it; 4Interdisciplinary Department of Medicine, University of Bari “Aldo Moro”, 70121 Bari, Italy; f.inchingolo@icloud.com; 5Multidisciplinary Department of Medical-Surgical and Dental Specialties, University of Campania “Luigi Vanvitelli”, Via Luigi de Crecchio 6, 80138 Naples, Italy; ludovica.nucci@unicampania.it

**Keywords:** kinesographic, muscular activity, orthodontics, palatal expansion, palatal hypoplasia, surgery, surgically assisted rapid palatal expansion, transcutaneous electrical nerve stimulation

## Abstract

The study aims to investigate the modifications in the temporalis and the masseter activity in adult patients before and after SARPE (Surgically Assisted Rapid Palatal Expansion) by measuring electromyographic and electrokinesographic activity. 24 adult patients with unilateral posterior crossbite on the right side were selected from the Orthodontic Department of the University of Milan. Three electromyographic and electrokinesographic surface readings were taken respectively before surgery (T0) and 8 months after surgery (T1). The electromyographic data of both right and left masseter and anterior temporalis muscles were recorded during multiple tests: standardized maximum voluntary contraction (MVC)s, after transcutaneous electrical nerve stimulation (TENS) and at rest. T0 and T1 values were compared with paired Student’s *t*-test (*p* < 0.05). Results: Significant differences were found in the activity of right masseter (*p* = 0.03) and right temporalis (*p* = 0.02) during clench, in the evaluation of right masseter at rest (*p* = 0.03), also the muscular activity of masseters at rest after TENS from T0 to T1 (pr = 0.04, pl = 0.04). No significant differences were found in the activity of left masseter (*p* = 0.41) and left temporalis (*p* = 0.39) during clench and MVC, in the evaluation of left masseter at rest (*p* = 0.57) and in the activity during MVC of right masseter (*p* = 0.41), left masseter (*p* = 0.34), right temporalis (*p* = 0.51) and left temporalis (*p* = 0.77). Results showed that the activity of the masseter and temporalis muscles increased significantly after SARPE during rest and clenching on the side where the cross-bite was treated.

## 1. Introduction

The main objective in orthodontics is to obtain oral health with a stable occlusion in harmony with all the functions of the stomatognathic system.

Maxillary transverse constriction is correlated to a variety of clinical sequelae. These include posterior cross-bite, crowded dentition, excessively wide buccal corridors and obstruction of nasal airway, which affects aesthetics, occlusal stability, atypical swallowing and the physiological stomatognathic functions. The treatment of choice for maxillary transverse constriction is the expansion of the median palatine suture in order to obtain a significant increase in maxillary width and dental arch perimeter.

Orthopedics rapid maxillary expansion (RME) is a procedure for the correction of maxillary contraction in children [1,2] and the surgically assisted rapid palatal expansion (SARPE) can be used on adult patients whose ossified sutures are resistant to orthopaedic forces [3]. The efficacy of SARPE as an expansion treatment on adults has been described [4,5,6]. This technique represents a suitable treatment for the correction of transverse maxillary deficiencies in adults, whereas the multi-segmental Le Fort I osteotomy is preferred in cases of transverse deficiencies associated with alterations on sagittal and vertical planes, thus solving the whole malocclusion in a single surgical procedure [7].

The evaluation and recording of the electrical activity produced by skeletal muscles (electromyography) and of the movement mechanisms (kinesiology) are fundamental to investigate the physiology of the facial musculature [8,9,10,11].

Other authors have investigated electromyographically the muscular activity of both the temporal and the masseter muscles in patients presenting cross-bite, concluding that, when a correct occlusal relationship is achieved, a normal muscle pattern can be obtained [12,13,14,15]. Other authors, instead, described that RME affects the muscular activity of the anterior temporalis and superficial masseter during the swallowing and chewing functions. Considering those factors, the variations in EMG activity should be examined in the treatment and its stability [16].

The activity of the masticatory muscles in the patients affected by crossbite has been described as asymmetric in the static [12,17] and dynamic phases [18,19,20]. On the other hand, different authors [21,22] described the asymmetry as non-statistically significant during rest, maximum teeth clenching and chewing.

The discrepancy might be due to the different methods and analysis described in the studies. Several studies described variability in the surface EMG (sEMG) in patients affected or unaffected by posterior cross-bite [2]. The biological variability, however, can affect the attempts to standardize the method [23]. sEMG standardization as a protocol is not sought only in crossbite patients, but is a reference for other muscles of the body [24].

De Rossi et al. described how the muscular activity changed, increasing in all the muscles examined after the expansion therapy compared with the pre-treatment analysis [25]. Arat et al. described how RME significantly decreased the EMG, and the activity of the anterior temporal and masseter muscles decreased significantly in the mono-lateral muscular activity of chewing [18]. Throckmorton et al. reported that the excursions during chewing resulted similarly in the patients affected by posterior crossbite who maintained the reverse chewing pattern [26]. The reverse sequence pattern can be described as a first medial deviation preceding the lateral excursion, to allow intercuspidation of the opposite cusps. Consequently the resolution of the crossbite should not change the central pattern that regulates the masticatory cycle [26]. Even though SARPE is becoming a very common practice, to date, no study seem to have analyzed the effects of the SARPE, and a sufficient number of studies assess the effects on the muscular activity before and after the regular palatal expansion.

The aim of this study is to investigate the changes produced by SARPE on neuromuscular activity.

## 2. Materials and Methods

The sample included 24 white patients (14 men and 10 women, mean age 30 ± 6 years) with unilateral posterior crossbite on the right side, participating in a prospective observational study on the muscular and kinesiographic activity after SARPE.

The study protocol was designed in accordance with the principles of the Declaration of Helsinki for medical research involving human subjects. All patients were informed on the characteristics of the study and agreed to participate by signing an EC-approved informed consent Institutional Review Board approval of the Ethics Committee of University of Milan (n. 421 10.09.2021).

Criteria for inclusion and exclusion in the study protocol are reported in Table 1.

All the recruited patients presented a transverse discrepancy of at least 5 mm measured in PAC (Postero Anterior Cephalometry) as first described by Ricketts. This differential indicates that a transverse discrepancy greater than Ricketts norm of 19.6 mm requires skeletal expansion, and that a surgical approach may need to be considered in adults [27]. Patients also presented a complete level of suture ossification according to their stage of growth as described by the Revelo and Fishman SMI (Skeletal maturation index) [28]. For this reason, maxillary expansion without surgery was not indicated.

SARPE was performed with Hyrax expander cemented the day before surgery.

The surgical technique involves the same sagittal maxillary surgery required for a LeFort I osteotomy, and included the separation of the pterygoid junction and of the mid-palatal suture between the roots of the incisors with an osteotome. During surgery, the device was activated to achieve a 2 to 2.5 mm separation of the maxillary central incisors. The same team led by an expert surgeon (more than 20 years of experience) performed all procedures.

After a latency period of 7 days, patients were instructed to activate the screw by 0.25 mm twice a day as described by Cureton et al. [29]. The patients were monitored once a week until the planned expansion was achieved as suggested by the same author, then the appliance was stabilized with a ligature wire. Active orthodontic treatment was initiated prior to SARPE in the mandibular arch (Figure 1).

The expansion device was left in place for approximately 8 months after the expansion.

The EMG activities of the anterior temporal and masseter muscles were recorded before SARPE (T0), and 8 months after surgery, at least 2 weeks after the removal of the expander (T1).

The electromyographic records were taken with two electromyographic devices: Freely electromyograph (De Gotzen, Legnano, Italy) (Figure 2) and K6-I electromyograph (Myotronics, Tukwila, WA, USA) (Figure 3).

Moreover, each subject was examined with a kinesiographic analysis of the mandibular excursion. No patient presenting facial, dental or skeletal pain was included prior to the analysis. The whole flowchart is visually represented in Figure 4.

The exams were conducted in a quiet room, without noise, with an ergonomic seat with a rigid stool, adjustable to the height of the patient until reaching 90° between the thighs and the lower leg, with the thighs parallel to the floor, and the head was positioned in the natural head position (NHP) [30].

Before positioning the electrodes, according to the manufacturer’s instructions the face was cleansed using Neoxinal (0.5% chlorhexidine in ≥70% alcohol), to prevent failure in the adhesion and to increase the adhesive effect. For both the intruments used the electrodes’ position was maintained.

We used disposable electrodes, bipolar-positioned according to the protocol procedure:Masseter muscle: The muscle mass was palpated by the operator from behind the chair and the patient was asked to perform clenching. To achieve parallelism between the muscle fibers and the bipolar electrode a line was traced on the skin from the labial commissure and the tragus.Temporal muscle: similarly, the muscle mass was palpated from behind during clenching activity to localize the major axis of the frontal bone zygomatic process. A line was traced parallel to the process and the electrode was placed.

The electromyographic measurements were analyzed with Freely.

The analysis was started during a maximum clench of 5 s performed on cotton rolls placed between the occlusal sides and then repeated without cotton rolls.

The protocol was designed as previously described by Ferrario et al. [15]. The K6-1 was used, firstly, to analyze the electromyographical activity during rest and to set the reference standard. For the mandibular activity, the following excursions were included: maximum opening; opening and closing; maximum mandible frontal protrusion; maximum left and right laterality; rest position of the mandible; centric occlusion; free airway at rest; free airway following transcutaneous electrical nerve stimulation (TENS); and finally, the distance between the maximum opening and the centric occlusion.

The mandibular EKG TENS was applied for a total of 45 min with the use of the two bipolar electrodes (Myotrode SG, and Myomonitor J5; Myotronics, Tukwila, WA, USA). Secondly, the EKG and EMG examinations were performed at rest, in order to evaluate the myo-activity reduction and the response to the SARPE.

The procedure was carried out twice, if any discrepancy higher than 5% was registered, the whole procedure was repeated until uniformity was achieved.

The mandibular movements during the swallowing activity were performed by swallowing water. The records registered showed a trace from the endpoint of deglutition to the maximum intercuspidation and described the vertical space between the occlusal plan in mm.

The EMG and EKG data collected at the different T of the treatment were statistically analyzed to determine the null hypothesis of showing alteration after the SARPE treatment.

### Statistical Evaluation

To evaluate the sample size we performed a one-sample mean *t*-test, with a power of 0.8 and alpha 0.05 and to obtain a standard deviation of 0.6 we obtained an estimated sample size of at least 20 patients. The statistical analysis was carried out with Student’s *t*-test (*p* < 0.05) performed by the use of the SPSS statistical software (IBM, Armonk, NY, USA) (Table 2). The null hypothesis assumes that the differences of values from T0 and T1 in the muscular activity were not reproducible; the alternative hypothesis, instead, would show different values in the two groups. Average and SD were calculated for each session.

## 3. Results

### 3.1. At Rest

Masseter muscle: overall mean rest activity after SARPE showed a higher increase on the right side (from T0 = 2.0 ± 0.6 μV to T1 = 3.1 ± 1.4 μV) rather than on the left side (from T0 = 1.9 ± 0.7 μV to T1 = 2.6 ± 1.7).

Activity at rest after transcutaneous electrical neural stimulation showed an increase, especially in the unaffected side (from T0 = 1.7 ± 0.5 μV to T1 = 2.9 ± 1.2 μV).

Temporalis muscle: overall mean activity at rest after SARPE increased more on the right side (from T0 = 2.1 ± 0.8 μV to T1 = 4.2 ± 2.2 μV) than on the left side (from T0 = 1.9 ± 0.7 μV to T1 = 2.6 ± 1.7).

Muscular activity at rest after TENS rose in both sides similarly, passing from T0 = 1.8 ± 0.6 μV to T1 = 3.4 ± 1.6 μV on the side with crossbite and from T0 = 3.0 ± 0.5 μV to T1 = 4.2 ± 1.7 μV on the contralateral side.

### 3.2. Maximum Clenching without Cotton Rolls

Masseter muscle: values of right masseter decreased considerably (from T0 = 67.1 ± 48.5 μV to T1 = 48.4 ± 31.4 μV). By contrast, on the side without crossbite, muscular activity increased after SARPE, passing from T0 = 49.3 ± 28.7 μV to T1 = 54.7 ± 32.9 μV.

Temporalis muscle: values of the right temporalis decreased (from T0 = 71.4 ± 47.4 μV to T1 = 56.3 ± 21.3 μV). Analogously, on the side without crossbite, muscular activity decreased after SARPE (from T0 = 75.0 ± 45.9 μV to T1 = 57.1 ± 35.2 μV).

### 3.3. Maximum Clenching with the Interposition of Cotton Rolls

Masseter muscle: values of the right masseter decreased (from T0 = 52.4 ± 36.2 μV to T1 = 48.9 ± 17.7 μV). By contrast, on the left side, muscular activity increased after SARPE (from T0 = 44.1 ± 18.8 μV to T1 = 52.4 ± 13.5 μV).

Temporalis muscle: values of the right temporalis decreased (from T0 = 59.5 ± 28.9 μV to T1 = 57.1 ± 22.9 μV). Analogously, on the unaffected side, muscular activity decreased after SARPE (from T0 = 68.7 ± 21.2 μV to T1 = 64.3 ± 15.4 μV).

Mean values, expressed in microvolts (µV) and standard deviations, are reported in Table 3.

Statistically significant differences (*p* = 0.03) from T0 to T1 on the values of the right masseter muscle at rest were found. The activity of the left masseter muscle at rest did not show a statistically significant difference from T0 to T1 (*p* = 0.57). The muscular activity of the right and the left masseter muscle at rest after TENS showed a statistically significant difference from T0 to T1 (pr = 0.04, pl = 0.04). During the maximum clench on cotton rolls, a statistically significant difference was not found in the activity of the right masseter muscle (*p* = 0.41), of the left masseter muscle (*p* = 0.34), of the right temporal muscle (*p* = 0.51) and in the activity of the left temporal muscle (*p* = 0.77). During the evaluation of maximum clench without cotton rolls, a statistically significant difference has been found in the activity of the right masseter muscle (*p* = 0.03) and the right temporal muscle (*p* = 0.02). No statistically difference has been found in the activity of the left masseter muscle (*p* = 0.41) and the left temporal muscle (*p* = 0.39) during clench. The maximum mouth opening increased after SARPE (T0 = 38.4 ± 1 mm; T1 = 40.9 ± 1.5 mm).

## 4. Discussion

The present study investigated the muscular activity of patients presenting unilateral skeletal crossbite before and after surgically assisted SARPE. The stomatognathic system is a functional and physiologic apparatus whose correct functioning relies on the equilibrium and balance of forces and structures [15]. A deficiency or excess on one side of the skull might cause a loss of equilibrium and develop a change in the muscular activity exacerbated during the growth of the subject. Normally, there is an adaptation or compensation to the skeletal alteration so the patient acquires new patterns to reestablish a correct function. One of the factors that might be influential is the occlusion, described as being dependent on muscular activity. Generally, when the occlusion is correct and in equilibrium, higher neuromuscular response during the muscular activity has been observed [31]. This is the reason why subjects with deviated masticatory patterns and altered occlusal positions have a major role in the topic. The muscular activity can be precisely measured with surface electromyographic instruments, compared to intramuscular devices [32,33]. The reproducibility of the instruments used for surface evaluation is well accepted in the literature and there are several standard protocols for their clinical usage [24,34].

It can be observed how the mean activity of masseter muscles in the side with crossbite during clench and during maximum cotton clench, specifically, the mean values of the right temporalis and left temporalis, decreased, as previously suggested by Arat et al. [18]. In fact, the author reported that RME significantly decreased the EMG activities of the anterior temporal and masseter muscles during chewing [18]. Other studies investigated the effects produced by the palatal expander on neuromuscular activity, for example De Rossi et al. [25] reported a significant increase in the masseter and temporalis muscles in adolescents, and, similar to the findings in this study, on adults. According to Michelotti et al. [35], the activity did not significantly became more symmetric, remaining asymmetric.

However, 1.5 months after the end of the expansion, the EMG activities of both muscles increased and reached pre-treatment values. Furthermore, the mean activity of masseter muscles both in the side with crossbite and in the contralateral side showed a statistically significant difference at rest after TENS between T0 and T1. Analogously, meaningful differences in the values of the right masseter activity at rest were found, suggesting muscular hyperactivity, as reported by Pinho et al. [35]. The authors found that EMG activity of the masticatory muscles during rest increased in patients with stomatognathic system dysfunctions compared with healthy subjects. This indicates an increase in basal tone [36]. Patients with maxillary hypoplasia can have reduced bite force of muscle function during chewing or clenching [37]. Furthermore, the SARPE expansion showed how the results can be compared to adult patients with analogous surgical procedures.

### Limitations

A few limitations of the study must be taken into account. First, the SARPE is not a frequent procedure, so the sample was limited. For this reason we could not include any patient presenting orofacial pain, swelling or any kind of sequelae after the intervention to eventually evaluate how it affected muscular activity, especially during the clenching activity. We could only analyze patients from the same ethnicity, and a longer follow-up was not considered. Future studies and implementations should take into account other different treatment procedures and longer follow ups.

## 5. Conclusions

Electromyographic analysis showed that the activity of the masseter and temporalis muscles increased significantly after SARPE during rest and clenching in the side affected by cross-bite in comparison with the side without cross-bite after SARPE.

## Figures and Tables

**Figure 1 bioengineering-09-00361-f001:**
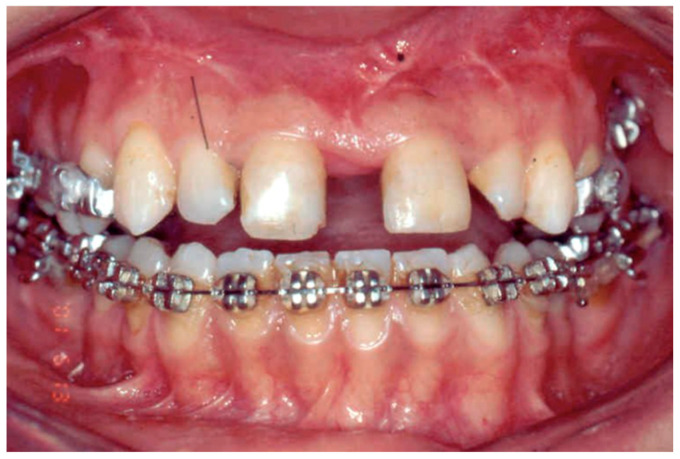
Sarpe activation and fixed theraphy initiated in the mandibular arch.

**Figure 2 bioengineering-09-00361-f002:**
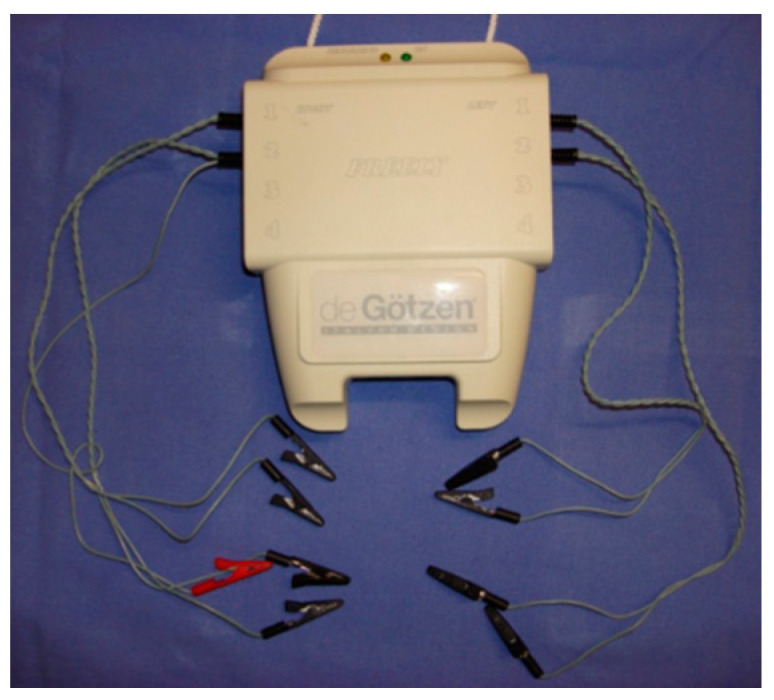
Freely electromyograph (De Gotzen, Legnano, Italy).

**Figure 3 bioengineering-09-00361-f003:**
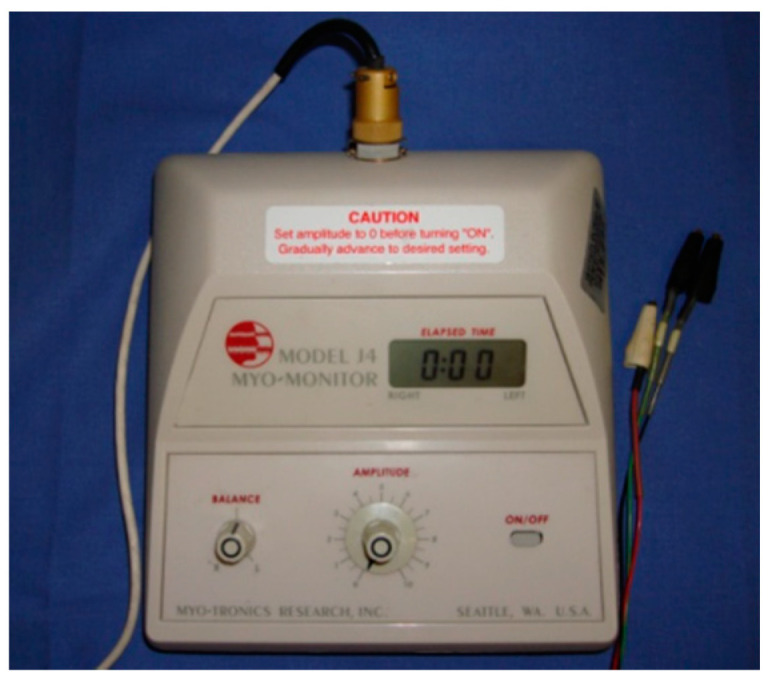
K6-I electromyograph (Myotronics, Tukwila, WA, USA).

**Figure 4 bioengineering-09-00361-f004:**
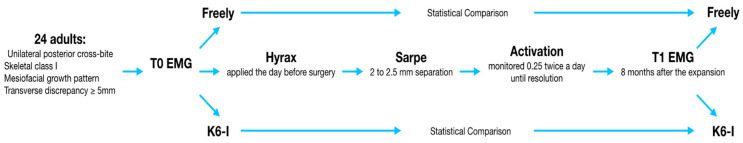
The flowchart of the study.

**Table 1 bioengineering-09-00361-t001:** Criteria for inclusion and exclusion in the study.

Inclusion Criteria	Exclusion Criteria
Skeletal class I	Craniofacial anomalies
Mesiofacial growth pattern	Temporomandibular joint dysfunction
Unilateral posterior cross-bite	Previous or current orthodontic treatment
Transverse discrepancy ≥ 5 mm	History of neuromuscular disease or disease affecting neuromuscular performance
	Diseases, syndromes, conditions as Vitamin deficiency or the habit of smoking and alcoholism.

**Table 2 bioengineering-09-00361-t002:** The statistical results of the T0/T1 sessions’ comparison. Legend: MM = masseter muscle; TA = temporal muscle; L = left; R = right; T0 = pre-treatment period; T1 = post-treatment period; AMR = muscular activity at rest; AMR TENS = muscular activity at rest after transcutaneous electrical neural stimulation; COTTON = force exerted on the maximum voluntary clenching on cotton rolls; CLENCH = force exerted on the maximum voluntary clenching on teeth.

Exam	T0; T1
AMR T0/T1 MM R	0.03
AMR T0/T1 MM L	0.57
AMR T0/T1 TA R	0.03
AMR T0/T1 TA L	0.79
COTTON T0/T1 MM R	0.41
CLENCH T0/T1 MM R	0.03
COTTON T0/T1 MM L	0.34
CLENCH T0/T1 MM L	0.41
COTTON T0/T1 TA R	0.51
CLENCH T0/T1 TA R	0.02
COTTON T0/T1 TA L	0.77
CLENCH T0/T1 TA L	0.39
AMR TENS T0/T1 MM R	0.04
AMR TENS T0/T1 MM L	0.04
AMR TENS T0/T1 TA R	0.25
AMR TENS T0/T1 TA L	0.34

**Table 3 bioengineering-09-00361-t003:** Mean values (µV) and standard deviations. Legend: MM = masseter muscle; TA = temporal muscle; L = left; R = right; T0 = pre-treatment period; T1 = post-treatment period; AMR = muscular activity at rest; AMR TENS = muscular activity at rest after transcutaneous electrical neural stimulation; COTTON = force exerted on the maximum voluntary clenching on cotton rolls; CLENCH = force exerted on the maximum voluntary clenching on teeth.

	MM R	MM L	TA R	TA L
AMR T0	2.0 ± 0.6	1.9 ± 0.7	2.1 ± 0.8	2.8 ± 0.5
AMR T1	3.1 ± 1.4	2.6 ± 1.7	4.2 ± 2.2	3.2 ± 1.7
AMR TENS T0	1.6 ± 0.6	1.7 ± 0.5	1.8 ± 0.6	3.0 ± 0.5
AMR TENS T1	2.1 ± 0.7	2.9 ± 1.2	3.4 ± 1.6	4.2 ±1.7
COTTON T0	52.4 ± 36.2	44.1 ± 18.8	59.5 ± 28.9	68.7 ± 21.2
COTTON T1	48.9 ± 17.7	52.4 ± 13.5	57.1 ± 22.9	64.3 ± 15.4
CLENCH T0	67.1 ± 48.5	49.3 ± 28.7	71.4 ± 47.4	75.0 ± 45.9
CLENCH T1	48.4 ± 31.4	54.7 ± 32.9	56.3 ± 21.3	57.1 ± 35.2

## Data Availability

No new data were created or analyzed in this study. Data sharing is not applicable to this article.

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
