# Peer review of "Effects on Muscular Activity after Surgically Assisted Rapid Palatal Expansion: A Prospective Observational Study"

_bioengineering, 2022, doi:10.3390/bioengineering9080361_

Round 1

Reviewer 1 Report

The manuscript investigates the effects on Muscular activity after surgically assisted rapid palatal expansion. However, the manuscript could be considered for publication after the revision.

1)    Table 2, caption for the two column is missing.

2)    2) Apart from the exclusion criteria taken into account, was any deficiency, person habits taken into consideration? For example, smoking, vit C deficiency.

3)    The author donot take into account any swelling behavior or pain as a result of surgery? If these are taken into account, then what effect would it have on the muscular activities while clenching?

4)    The author states that the patients were monitored once a week until the planned expansion was achieved. Was the expansion consistent throughout the process? Did it result in any swelling behavior in the patient and if so, then what was the factors taken into considerations.

5)    There is no pictographically representation, it would be good have some in the main text.

6)    What was the protocol followed to capture the electromyographical signals?

7)    Is there any analysis conducted for looking into the influence of longer postoperative period?

Author Response

Reviewer #1

The manuscript investigates the effects on Muscular activity after surgically assisted rapid palatal expansion. However, the manuscript could be considered for publication after the revision.

Dear reviewer, we are very grateful for your kind suggestions and comments, please find our responses below after each one of the points you gave us to increase the quality of our manuscript. We would be very grateful if you could kindly express a feedback on the review made if every point was correctly answered and modified accordingly.

1)    Table 2, caption for the two column is missing.

Dear reviewer, thanks, we added the captions for each column of table II.

2)    2) Apart from the exclusion criteria taken into account, was any deficiency, person habits taken into consideration? For example, smoking, vit C deficiency.

You are right, now it is described more accurately in the proper section and in Table I among the exclusion criteria, we decided also to add to the exclusion criteria any patient which has  alcoholism or any other syndrome/condition.

3)    The author do not take into account any swelling behavior or pain as a result of surgery? If these are taken into account, then what effect would it have on the muscular activities while clenching?

In the sample observed, after a period of 8 months after surgery, we did not have any adverse effect or sequelae of the surgery as swelling of the area or pain, of course this is a limitation that must be taken into account, we added a paragraph explaining our decision, kindly let us know if you think it was appropriately addressed.

4)    The author states that the patients were monitored once a week until the planned expansion was achieved. Was the expansion consistent throughout the process? Did it result in any swelling behavior in the patient and if so, then what was the factors taken into considerations.

Dear reviewer, we applied thoroughly the guidelines and the protocol described by Cureton et al, precisely: “Do not activate appliance until the gingiva is healthy.” and “Activate the appliance one turn every day and evaluate weekly.” and we did not found relevant swelling or counter indications, it is now specified through the text.

5)    There is no pictographically representation, it would be good have some in the main text.

Dear reviewer, we added clinical photos, kindy let us know if they are helpful.

6)    What was the protocol followed to capture the electromyographical signals?

Dear reviewer, we are not sure if we understand correctly your question, the protocol was designed as previously described by Ferrario et al. and it is described in the materials and methods section from: “The exams were conducted in a quiet room, without noises, with a ergonomic seat…” to the end of the paragraph. We also design the protocol according to the producer Company’s instructions.

7)    Is there any analysis conducted for looking into the influence of longer postoperative period?

We could not perform follow ups for more than 8 months without expecting some drop outs, for this reason we did exceeded that time span, however this was added as a limitation.

Reviewer 2 Report

The topic of the manuscript is good and clinically relevant.

However, there are some shortcomings that make it unacceptable for publication.

1. The manuscript lacks the control category. The experimental design is flawed.

2. Is the number of samples for the experiment sufficient?

3. Tables 2 and 3 alone hardly support the conclusions of this manuscript.

4. It is recommended to supplement the group and sample size. Also, consider what analysis needs to be added to support the conclusions of this study.

“The aim of this study is to investigate the changes produced by SARPE on neuromuscular activity.”

It is recommended to highlight the importance of the topic.

Author Response

Reviewer #2

The topic of the manuscript is good and clinically relevant.

However, there are some shortcomings that make it unacceptable for publication.

Dear reviewer, thanks for taking your time to express your opinion on our manuscript, we understand that we might have introduced some flaws in our study, however, considering the fact that this is the first study analyzing the muscular activity after a surgically assisted expansion procedure we kindly ask if you would like to look again at the manuscript and see if the modifications introduced might have increased the quality. Please also keep in mind that other studies in the field of muscular activity after expansion usually present the same design, style and sample, please have a look at the following studies:
https://doi.org/10.1016/j.ajodo.2006.07.028 and Neuromuscular evaluation in young patients with unilateral posterior crossbite before and after rapid maxillary expansion. 17(3), 84–88.

  1. The manuscript lacks the control category.The experimental design is flawed.
  2. Is the number of samples for the experiment sufficient?

Dear reviewer, thanks for your question, we statistically calculated the estimated sample size, you can find it under materials and methods now, kindly tell us if this answers your question. Also this is coherent with previous works with a high impact on the literature, for example: https://doi.org/10.1016/j.ajodo.2006.07.028 18 cases 10.1093/ejo/cjy019 29 cases https://doi.org/10.1016/j.ajodo.2007.08.027 27 cases

  1. Tables 2 and 3 alone hardly support the conclusions of this manuscript.

Dear reviewer, table 2 and 3 report the statistical analysis between t0 and t1 of different Exams, we did our best to provide the readers a simple, synthesized version of an otherwise big amount of data.

  1. It is recommended to supplement the group and sample size. Also, consider what analysis needs to be added to support the conclusions of this study.

The aim of this study is to investigate the changes produced by SARPE on neuromuscular activity.”

It is recommended to highlight the importance of the topic.

Dear reviewer, as previously stated, we added the sample size, and and the highlight for the importance of the study, the aim is provided by analyzing the muscular activity before and after the treatment, kindly tell us if the modifications were provided correctly.

Reviewer 3 Report

I would like to congratulate the authors for conducting the present study. Here goes a few suggestions:

Please remove the subheading from the Abstract. According to the journal guidelines they are not required

Please explain the meaning of TENS in the Abstract

I recommend the authors to place the keywords by alphabetic order

Try not o use abbreviations in the keywords such as TENS

Why was the ethnic group an inclusion criteria?

In the discussion there is no debate regarding the comparison of the present research with previous studies. Can that be added?

May the authors debate the study strength and limitations

May the authors debate the generalization of the results and future studies perspectives.

Please notice that multiple references in the reference list are not according to the authors guidelines.

Author Response

Reviewer #3

I would like to congratulate the authors for conducting the present study. Here goes a few suggestions:

Dear reviewer, we are very thankful for your kind comment and for using your time to give us excellent comments to increase the quality of our manuscript and to correct the errors, please find our comments and answers below.

Please remove the subheading from the Abstract. According to the journal guidelines they are not required

Dear reviewer we removed “the subheading from the abstract

Please explain the meaning of TENS in the Abstract

I recommend the authors to place the keywords by alphabetic order

Try not o use abbreviations in the keywords such as TENS

Dear reviewer, we extended TENS in the abstract and among the keywords. And we fixed the word order.

Why was the ethnic group an inclusion criteria?

Dear reviewer, we removed the ethnic group as inclusion criteria, we wanted to specify that all the patients were from the same ethnic group, it is now added throughout the text.

In the discussion there is no debate regarding the comparison of the present research with previous studies. Can that be added?

May the authors debate the study strength and limitations

Dear reviewer we added a whole new limitation section, kindly revise if there is any limitation we might have forgotten.

May the authors debate the generalization of the results and future studies perspectives.

We added a few sentence by the end of the discussion section.

Please notice that multiple references in the reference list are not according to the authors guidelines.

Dear reviewer, we fixed the discrepancies in the reference formatting according to the Journal’s style, thanks.

Round 2

Reviewer 1 Report

The authors have satisfactorily responded to all my questions and made the necessary changes to the manuscript. 
I recommend the manuscript in its present form.

Author Response

Dear reviewer, we are grateful for taking the time to give us your opinion.

Reviewer 2 Report

1. I suggest supplementing the previous literature.

2. In addition, a flowchart of this study has been added.

3. Also, how to prove that this study is to be believed? (very little data)

4. “Dear reviewer, as previously stated, we added the sample size, and and the highlight for the importance of the study, the aim is provided by analyzing the muscular activity before and after the treatment, kindly tell us if the modifications were provided correctly.

I want you to be able to articulate the impact between muscle and corrective treatments. Because some readers are not experts in this area.

Author Response

1. I suggest supplementing the previous literature.

Dear reviewer, we added a few paper from previously published literature.

2. In addition, a flowchart of this study has been added.

Dear reviewer, we added a flowchart at the end of the study description, as figure 4.

3. Also, how to prove that this study is to be believed? (very little data)

Dear reviewer, we believe in the methods used previously described by a multitude of previously published literature, we believe in the validity of the instruments used as these underwent certification procedures, we do not believe in our null hypothesis, and that’s why we applied statistics to validate them, under the limitations of the study.

4. “Dear reviewer, as previously stated, we added the sample size, and and the highlight for the importance of the study, the aim is provided by analyzing the muscular activity before and after the treatment, kindly tell us if the modifications were provided correctly.”

I want you to be able to articulate the impact between muscle and corrective treatments. Because some readers are not experts in this area. 

Dear reviewer, we are grateful for this comment, we modified the conclusion accordingly,

Reviewer 3 Report

Dear authors. I have no more concerns. Thank you. 

Author Response

Dear reviewer, Thank you for helping us improving our manuscript.

This manuscript is a resubmission of an earlier submission. The following is a list of the peer review reports and author responses from that submission.